# Platelets Inhibit Methicillin-Resistant *Staphylococcus aureus* by Inducing Hydroxyl Radical-Mediated Apoptosis-Like Cell Death

Erxiong Liu,[a] Yutong Chen,[a] Jinmei Xu,[a] Shunli Gu,[a] Ning An,[a] Jiajia Xin,[a] Wenting Wang,[a] Zhixin Liu,[a] Qunxing An,[a] Jing Yi,[a] Wen Yin[a]

[a]Department of Transfusion Medicine, Xijing Hospital, Fourth Military Medical University, Xi'an, Shanxi, China

Erxiong Liu, Yutong Chen, and Jinmei Xu contributed equally to this work. Author order was determined by the corresponding authors after negotiation.

**ABSTRACT** Methicillin-resistant *Staphylococcus aureus* (MRSA) is one of the most common drug-resistant bacteria and poses a significant threat to human health. Due to the emergence of multidrug resistance, limited drugs are available for the treatment of MRSA infections. In recent years, platelets have been reported to play important roles in inflammation and immune responses, in addition to their functions in blood hemostasis and clotting. We and other researchers have previously reported that platelets can inhibit *Staphylococcus aureus* growth. However, it remained unclear whether platelets have the same antibacterial effect on drug-resistant strains. In this study, we hypothesized that platelets may also inhibit the growth of MRSA; the results confirmed that platelets significantly inhibited the growth of MRSA *in vitro*. In a murine model of MRSA infection, we found that a platelet transfusion alleviated the symptoms of MRSA infection; in contrast, depletion of platelets aggravated infective symptoms. Moreover, we observed an overproduction of hydroxyl radicals in MRSA following platelet treatment, which induced apoptosis-like death of MRSA. Our findings demonstrate that platelets can inhibit MRSA growth by promoting the overproduction of hydroxyl radicals and inducing apoptosis-like death.

**IMPORTANCE** The widespread use of antibiotics has led to the emergence of drug-resistant bacteria, particularly multidrug-resistant bacteria. MRSA is the most common drug-resistant bacterium that causes suppurative infections in humans. As only a limited number of drugs are available to treat the infections caused by drug-resistant pathogens, it is imperative to develop novel and effective biological agents for treating MRSA infections. This is the first study to show that platelets can inhibit MRSA growth *in vitro* and *in vivo*. Our results revealed that platelets enhanced the production of hydroxyl radicals in MRSA, which induced a series of apoptosis hallmarks in MRSA, including DNA fragmentation, chromosome condensation, phosphatidylserine exposure, membrane potential depolarization, and increased intracellular caspase activity. These findings may further our understanding of platelet function.

**KEYWORDS** platelets, MRSA, hydroxyl radical, apoptosis-like death

The emergence and spread of multidrug-resistant bacteria have become a global medical challenge. According to the World health Organization, 700,000 people die from multidrug-resistant "superbugs" every year, and the number is estimated to reach 10 million by 2050 (1–4). MRSA is a pathogenic bacterium responsible for a large number of nosocomial infections; MRSA ranks first among the three major pathogenic microorganisms prevalent worldwide (*Mycobacterium tuberculosis*, human immunodeficiency virus, and Methicillin-resistant *Staphylococcus aureus* [MRSA]) (5) and accounts for 20% to 50% of clinically isolated *S. aureus* in many regions of the world (6, 7). Clinical studies have shown that vancomycin efficacy is decreasing, and the emergence

Address correspondence to Wen Yin, yinwen@fmmu.edu.cn.

The authors declare no conflict of interest.

of multidrug-resistant "super bacteria" has added new challenges to the treatment of MRSA infection (2, 8–10). Therefore, it is imperative to develop new antibacterial agents and infection control strategies.

Recent studies have confirmed that platelets play important roles in immune defence. Platelets monitor the integrity of blood vessels and help the body build an effective immune system (11). During infection, platelets quickly travel to the site of infection in large numbers (12). Platelets express several toll-like receptors (TLRs) such as TLR-2, TLR-4, and TLR-9, which can recognize and interact with pathogens and eventually result in platelet activation and aggregation (13–15). Activated platelets exert antibacterial effects directly through antimicrobial peptides (AMPs) secreted by $\alpha$-particles (16–19). Moreover, platelet-derived kinocidins can recruit, activate, and promote other immune cells, including neutrophils, macrophages, and dendritic cells, to play antibacterial roles in addition to directly killing pathogens (20–24). Studies have reported that platelets transport blood-borne bacteria to the CD8$\alpha^+$ dendritic cells via glycoprotein GPIb and complement C3 (25). The migrating platelets collect bacteria as scavengers and facilitate neutrophil activation during sepsis (26). Platelet-derived chemokine (C-X-C motif) ligand (CXCL7) can be cleaved to form four different chemokine peptides, namely, PBP, CTAP-III, $\beta$-TG, and NAP-2 (27). The C-terminal truncation of NAP-2 produces two other peptides, namely, thrombin 1 and 2, which exert direct bactericidal effects on *Bacillus subtilis*, *S. aureus*, and *Cryptococcus neoformococcus in vitro* (28). In addition, we previously confirmed that platelets directly inhibit *S. aureus* by secreting TGF-$\beta$1 (29). However, it remained unclear whether platelets have the same antibacterial effect on drug-resistant strains. MRSA is one of the most common multidrug-resistant bacteria, which poses a significant threat to human health (1, 4). It is, therefore, very meaningful to study the antibacterial effect of platelets on MRSA. Based on available scientific evidence, we hypothesized that platelets may inhibit MRSA growth as well as the mechanisms underlying their infection.

In the present study, we confirmed the inhibitory effect of platelets on MRSA growth *in vitro* and *in vivo*. Treatment with platelets induced the overproduction of hydroxyl radicals in MRSA, which resulted in apoptosis-like death (ALD) of MRSA. Our findings provide important insights into the functions of platelets beyond blood homeostasis. Moreover, this study provides new insights into the antibacterial function of platelets.

## RESULTS

**Platelets directly inhibit MRSA growth *in vitro*.** To detect the direct antibacterial effect of platelets on MRSA, we cocultured purified platelets with MRSA. First, we confirmed the drug resistance of MRSA standard strains (USA300, ATCC BAA-1717) and clinical isolate strains (Fig. S1A to I). Then, we cocultured varying amounts of purified platelets with MRSA to establish the optimal ratio of platelets against MRSA (Fig. S1J and K). We observed that turbidity of the PLT-MRSA group (indicating MRSA growth) was much lower than that of the MRSA group (Fig. 1A). Moreover, the optical density at 600 nm (OD$_{600}$) decreased severely after 10 h of incubation with platelets (Fig. 1B). The clinical isolates of MRSA cocultured with platelets showed results similar to the standard strains (Fig. S2A to C). Moreover, several dead MRSA were found to be surrounded by lysed platelets (Fig. S2D). We detected MRSA division and proliferation using carboxyfluorescein succinimidyl ester (CFSE) and found that the fluorescence intensity of carboxyfluorescein did not decrease in the PLT-MRSA group, indicating that division and proliferation of MRSA were inhibited following platelet treatment (Fig. 1C and D). Based on the OD$_{600}$ value and bacterial colony counts of MRSA, we found that the platelet-mediated inhibition of MRSA started after 6 h of coculture with platelets, reached the maximum at 10 h, and lasted for 24 h (Fig. 1E to H). These results demonstrate that platelets directly inhibit MRSA growth *in vitro*.

**Platelets inhibit MRSA growth *in vivo*.** To confirm our findings, we established a murine model of MRSA infection. We observed that body weight and the number of

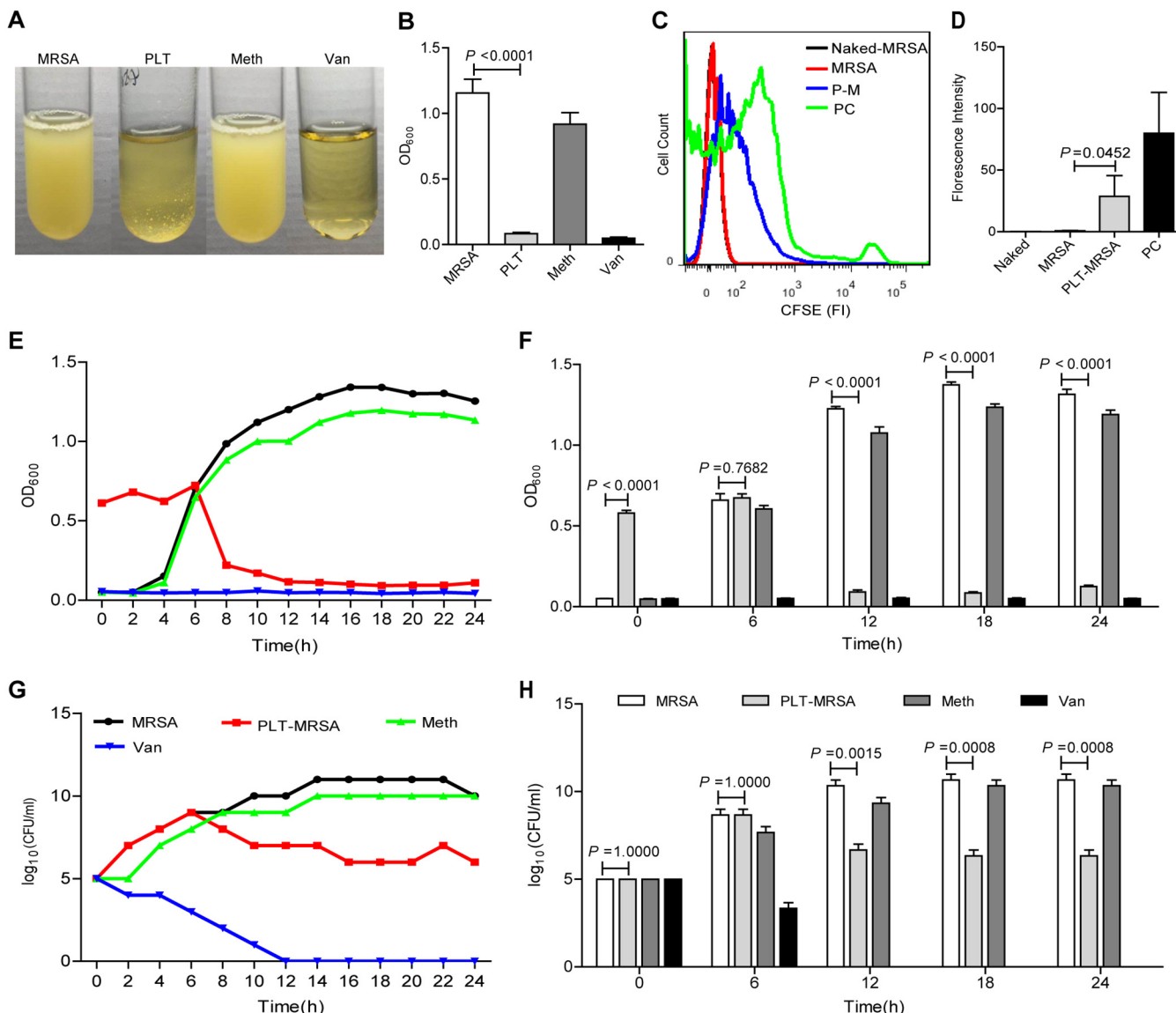

**FIG 1** Platelets inhibit MRSA growth *in vitro*. MRSA (USA300) were cocultured with or without platelets for 10 h. MRSA, untreated, as a control; PLT, MRSA co-cultured with platelets; Meth, MRSA treated by methicillin (6.83 mg/L), as a negative control; Van, MRSA treated by vancomycin (4 mg/L), as a positive control. (A) The photos of MRSA in each group. (B) Comparison of $OD_{600}$ of each group. (C) Proliferation detection of MRSA using CFSE by flow cytometry. Naked-MRSA, unstained MRSA, as negative control; MRSA, untreated MRSA; PLT-MRSA, MRSA cocultured with platelets; PC, uncultured but stained MRSA, as positive control. (D) Statistical results of CFSE tests in (C). (E) The growth curve of MRSA cocultured with or without platelets for 24 h, according to $OD_{600}$. (F) $OD_{600}$ of MRSA at 0 h, 6 h, 12 h, 18 h, and 24 h after coculture. (G) The growth curve of MRSA cocultured with or without platelets for 24 h, according to bacteria counts. (H) Bacteria count at 0 h, 6 h, 12 h, 18 h, and 24 h after coculture. All results have been tested at least three times. Data presented as mean ± SEM. Student's *t* test for two-group comparisons.

white blood cells (WBCs) and platelets in mice decreased with an increase in MRSA count, and survival decreased after mice were injected with high numbers of MRSA (Fig. S3A to D). Significantly large numbers of MRSA were present in the peripheral blood of infected mice compared with those in the blood of the control mice (Fig. S3E). In addition, more leukocytes infiltrated the livers of MRSA-infected mice than those of the control mice, which suggested that the liver was seriously damaged by MRSA (Fig. S3F). These results indicated that the MRSA infection model was successfully established.

To verify whether platelets inhibited MRSA growth in mice, we freshly purified platelets from whole blood samples obtained from mice (Fig. S3G) and transfused them into the MRSA-infected mice (Fig. 2A). No difference was observed in body weight or WBC counts between the PLT-MRSA and MRSA groups (Fig. 2B and D). Platelet counts

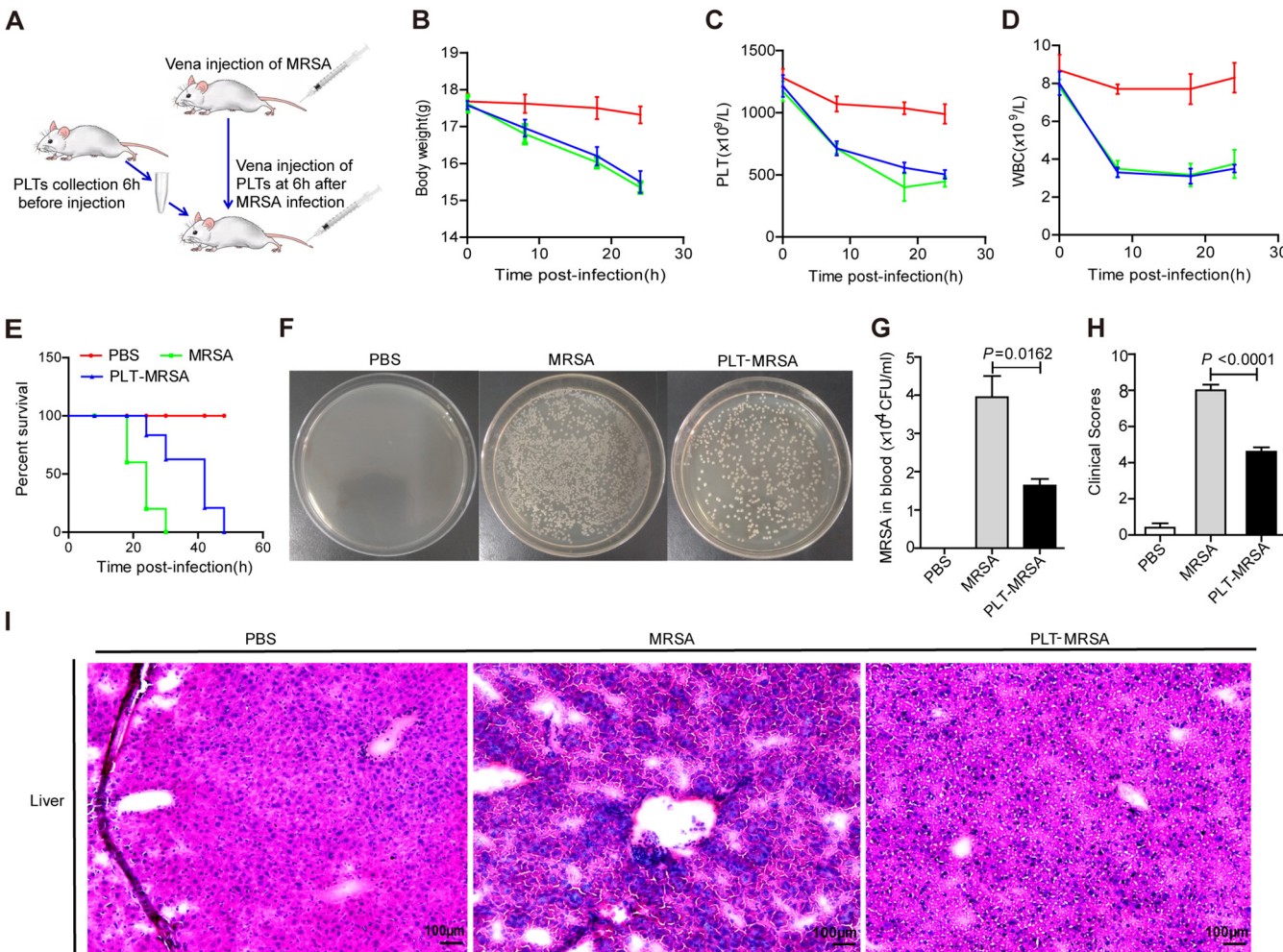

**FIG 2** Platelets transfusion alleviate the symptoms of MRSA infection in mice. C57BL/6 mice were infected with MRSA, then administered platelets transfusion or not (*n* = 5 mice/group; 6 to 8 weeks old, female). PBS, mice were injected with PBS, as negative control group; MRSA, mice were infected with MRSA. PLT-MRSA, mice were injected with platelets after MRSA infection. (A) Schematic diagram of platelets transfusion in MRSA infected mice. (B) Changes of body weight of mice in each group. (C) Peripheral blood platelet counts of mice in each group. (D) Peripheral blood WBC counts of mice in each group. (E) Survival curves of MRSA infected mice in each group. (F) The amount of MRSA in peripheral blood of infected mice in each group after 24 h. (G) Statistical histogram of bacteria counts in (F). (H) Clinical score of mice status according to the grade assessment. (I) Representative images of H&E staining of liver. The original magnification was 200× in each group. All results have been tested at least three times. Data presented as mean ± SEM. Student's *t* test for two-group comparisons.

in the PLT-MRSA group increased after platelet transfusion (Fig. 2C), indicating that platelet transfusion was successful. Furthermore, survival was prolonged in the PLT-MRSA group (Fig. 2E), and the number of MRSA in the peripheral blood decreased after 24 h (Fig. 2F and G). Moreover, the infection symptoms were significantly alleviated after platelet transfusion (Fig. 2H) and the damage to the liver was also relieved to some extent (Fig. 2I). These data suggest that platelet transfusion can inhibit MRSA growth *in vivo*.

To further validate the direct effect of platelets against MRSA *in vivo*, we used an anti-CD42b monoclonal antibody (MAb) to deplete platelets in mice; the mice were then injected with MRSA 2 h later (Fig. 3A). The body weight and platelet counts decreased in the PLT-depleted mice, although the WBC counts increased (Fig. 3B to D). Furthermore, the PLT-depleted mice showed reduced survival and worse clinical scores (Fig. 3E and H), as their peripheral blood MRSA count increased significantly after 18 h of infection and liver damage was also aggravated (Fig. 3F, G, and I). These results suggest that platelet depletion aggravates MRSA infection symptoms

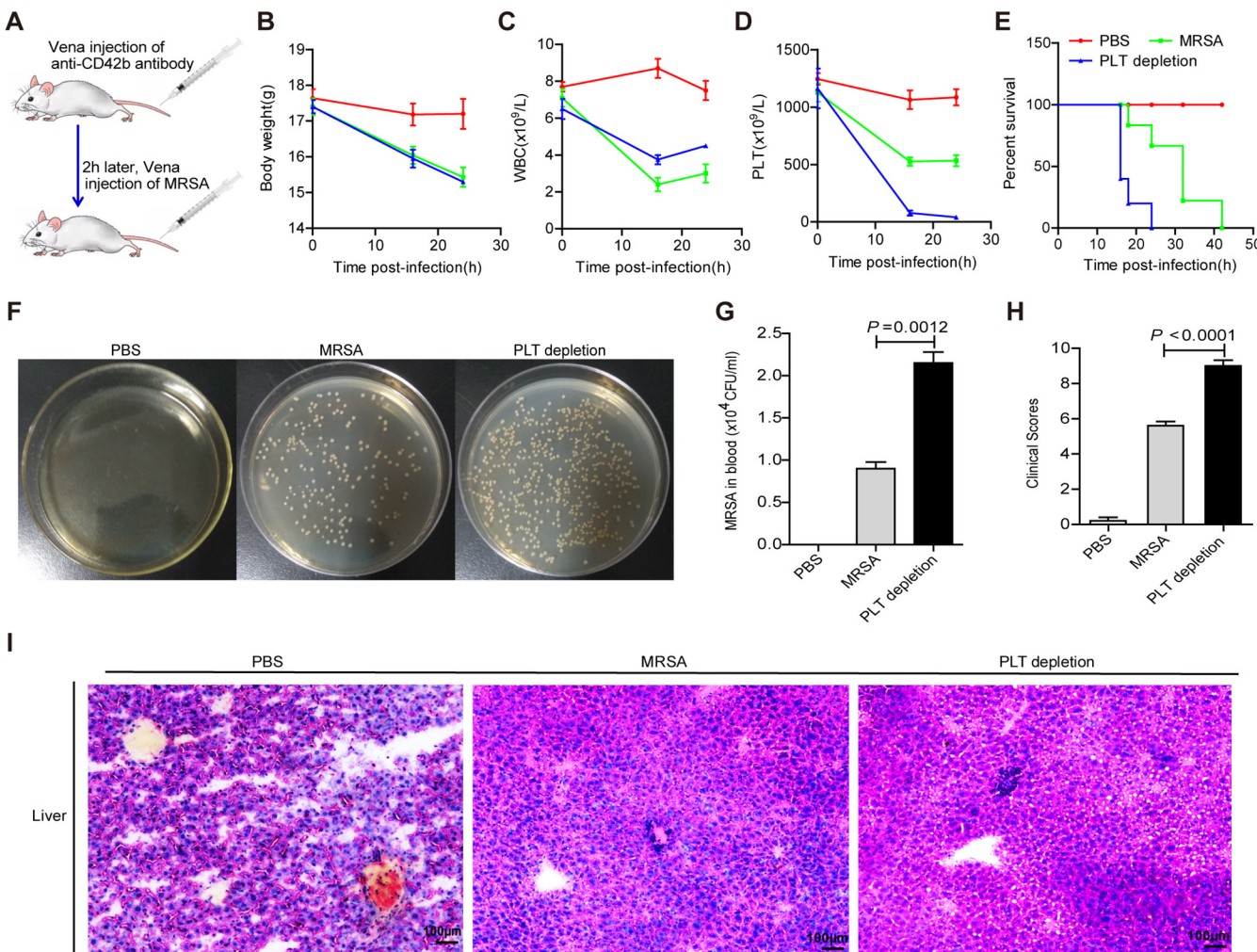

**FIG 3** Depletion of platelets aggravate the symptoms of MRSA infected mice. Murine platelets were exhausted with anti-CD42b MAb 2 h before MRSA injection (*n* = 5 mice/group; 6 to 8 weeks, female). PBS, control group; MRSA, mice were infected with MRSA. PLT depletion, murine platelets were eliminated by MAb and then infected with MRSA. (A) PLT depletion protocol in MRSA infected mice. (B) Changes in body weight of mice. (C) Counts of WBC in peripheral blood. (D) Numbers of platelet. (E) Survival curve of mice. (F) The amount of MRSA in peripheral blood at 24 h after infection. (G) Quantification of bacteria counts in (F). (H) Clinical score of mice status according to the grade assessment. (I) Representative images of H&E staining of liver. The original magnification was 200× in each group. All results have been tested at least three times. Data presented as mean ± SEM. Student's *t* test for two-group comparisons.

in mice. Taken together, these data demonstrate that platelets can directly inhibit MRSA *in vivo*.

**Platelets inhibit MRSA growth by inducing excessive production of hydroxyl radicals.** As our results showed that platelets exert inhibitory effects on MRSA *in vivo* and *in vitro*, we next aimed to elucidate the mechanism underlying the platelet-mediated inhibition of MRSA growth. In our previous study, we performed RNA sequencing of *S. aureus* after platelet treatment (29). We found that platelet treatment modulated the transcriptional levels of the components involved in the oxidative phosphorylation signalling pathway in *S. aureus* (Fig. S4A). Cluster analyses of the RNA sequencing data showed that expression of the NADH dehydrogenase-related gene *ndhF* and the cytochrome c oxidase-related genes *qoxB*, *qoxC*, and *qoxD* of *S. aureus* was upregulated (Fig. S4B). Here, we determined the changes in the expression of these genes in MRSA after platelet treatment. We found that the expression of *ndhO*, *ndhR*, *qoxC*, and *qoxD* was upregulated (Fig. 4A). Expression of the *azr* gene, which is responsible for providing electrons to the respiratory chain, was downregulated (Fig. 4A). A reduction in AZR expression can compromise electron supply to the oxidative respiratory chain (30). The increase in NADH dehydrogenase NadhF and cytochrome c oxidase QoxB, QoxC, and

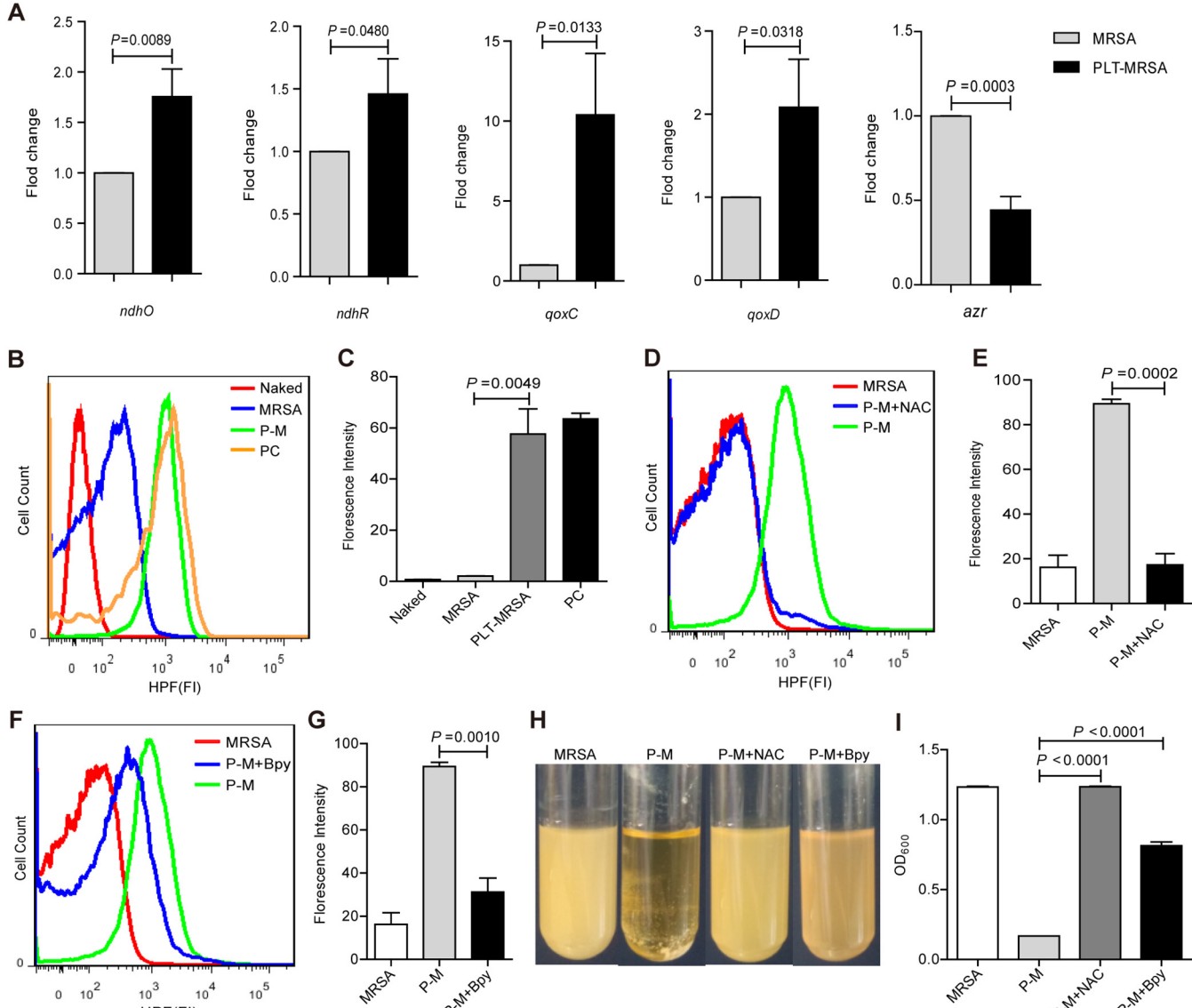

**FIG 4** Platelets induce oxidative stress and OH• overproduction in MRSA. MRSA were cocultured with or without platelets for 10 h, adding 6 mM NAC and 500 $\mu$M Bpy, respectively, to the coculture system. Naked-MRSA, unstained MRSA; MRSA, untreated MRSA; P-M, platelets treated MRSA; P-M+NAC, NAC was added into the platelets treated MRSA system. P-M+Bpy, Bpy is added to platelets treated MRSA system. (A) Validation of RNA-sequence results by qRT-PCR. (B) The formation of OH• detected by flow cytometry. PC, MRSA treated with MMC (5 $\mu$g/mL) for 4 h, as a positive control. (C) Statistical results of FITC-HPF fluorescence intensity in (B). (D) The formation of OH• after adding NAC. (E) Statistical results of fluorescence intensity in (D). (F) The formation of OH• after adding Bpy. (G) Statistical results of fluorescence intensity in (F). (H) Bacterial fluid turbidity of each group. (I) $OD_{600}$ detection of each group in (H). All results have been tested at least three times. Data presented as mean $\pm$ SEM. Student's $t$ test for two-group comparisons.

QoxD expression suggested a further reduction in the electron supply to the oxidative respiratory chain (31, 32). These results provided evidence that MRSA suffered severe oxidative stress following platelet treatment.

In general, reactive oxygen species (ROS) consist of three main types: superoxide ($O_2$-), hydrogen ($H_2O_2$), and hydroxyl radicals (OH•) (33). When cells are subjected to oxidative stress, $O_2$- produced by the oxidative respiration chain is converted into $H_2O_2$ under the action of superoxide dismutase. $H_2O_2$ in the presence of $Fe^{2+}$ generates OH• through the Fenton reaction (33–35). Therefore, both $O_2$- and $H_2O_2$ can eventually be converted into OH• *in vivo*. OH• is considered the most toxic and deadly of the three types. The highly destructive OH• acts as an "executor" of cell death, which directly damages DNA, lipids, and proteins, and ultimately leads to cell death (36). Therefore, we used hydroxyphenyl fluorescein (HPF) to detect the production of OH• in MRSA. As

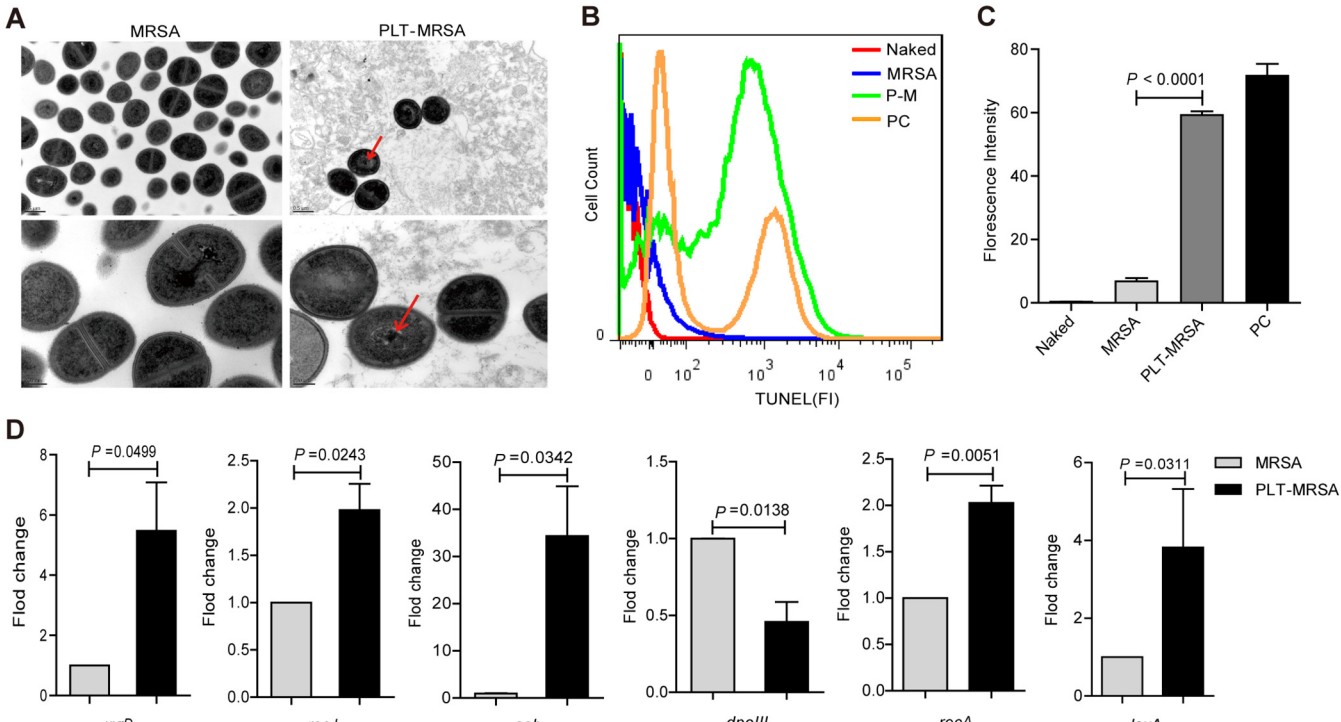

**FIG 5** Platelets induce chromatin condensation and DNA fragmentation of MRSA. MRSA were cocultured with or without platelets for 10 h. Naked, unstained MRSA; MRSA, untreated MRSA, as a control; PLT-MRSA, Platelets treated MRSA. (A) The ultrastructure of MRSA by transmission electron microscopy. Scale bars: 0.5 $\mu$m (upper panel); 200 nm (lower panel). The original magnification was 30,000$\times$ (upper panel), 80,000$\times$ (lower panel) in each group. (B) TUNEL detection of DNA damage in MRSA by FACS analysis. PC, MRSA treated with MMC (5 $\mu$g/mL) for 4 h, as a positive control. (C) Statistical results of fluorescence intensity in (B). (D) The detection of DNA damage repair related genes by $q$RT-PCR. All results have been tested at least three times. Data presented as mean $\pm$ SEM. Student's $t$ test for two-group comparisons.

shown in Fig. 4B and C, the fluorescence intensity of FITC-HPF significantly increased in the PLT-MRSA group, indicating that platelet treatment led to the overproduction of OH$^\bullet$ in MRSA. To confirm our findings, we used an OH$^\bullet$ scavenger N-acetylcysteine (NAC), which protects cells against oxidative stress and plays an important role in preventing DNA damage and regulating apoptosis, and a $Fe^{2+}$ chelator, 2,2'-bipyridine (Bpy), which inhibits OH$^\bullet$ generation by chelating $Fe^{2+}$ ions, a key component of the Fenton reaction (37). Addition of NAC and Bpy to the platelet and MRSA coculture led to a significant reduction in intracellular OH$^\bullet$ levels in MRSA (Fig. 4D to G). As a result, the turbidity of the bacterial suspension, as well as $OD_{600,}$ significantly increased, indicating that the platelet-mediated inhibitory effect on MRSA was weakened by the quenching of OH$^\bullet$ (Fig. 4H and I). These results imply that platelets inhibit MRSA growth by inducing OH$^\bullet$ overproduction. To confirm that the overproduced OH$^\bullet$ originated from MRSA rather than the platelets, we prepared high-purity platelet lysates (Fig. S5A and B) and evaluated their effect on MRSA proliferation (Fig. S5C to E). Significantly increased OH$^\bullet$ levels were observed in MRSA after platelet lysate treatment (Fig. S5F and G). Addition of NAC or Bpy to the platelet lysates and MRSA coculture significantly reduced the intracellular OH$^\bullet$ levels in MRSA (Fig. S5H to K) as well as the bacteriostatic effect of platelet lysates (Fig. S5L and M). These results suggest that platelet lysates can also inhibit MRSA growth by inducing OH$^\bullet$ overproduction.

**Platelets induce ALD of MRSA.** We previously reported that platelet treatment induces severe DNA damage in *S. aureus* (29). Therefore, we speculated that platelets may also damage the DNA of MRSA. First, we performed transmission electron microscopy to observe the changes in the morphology and internal structure of MRSA following platelet treatment. We observed notable condensation of MRSA chromatin after 10 h of platelet treatment, whereas no significant changes were observed in the cell wall and membrane (Fig. 5A). This indicated that platelet treatment destroyed the nuclear structure of MRSA.

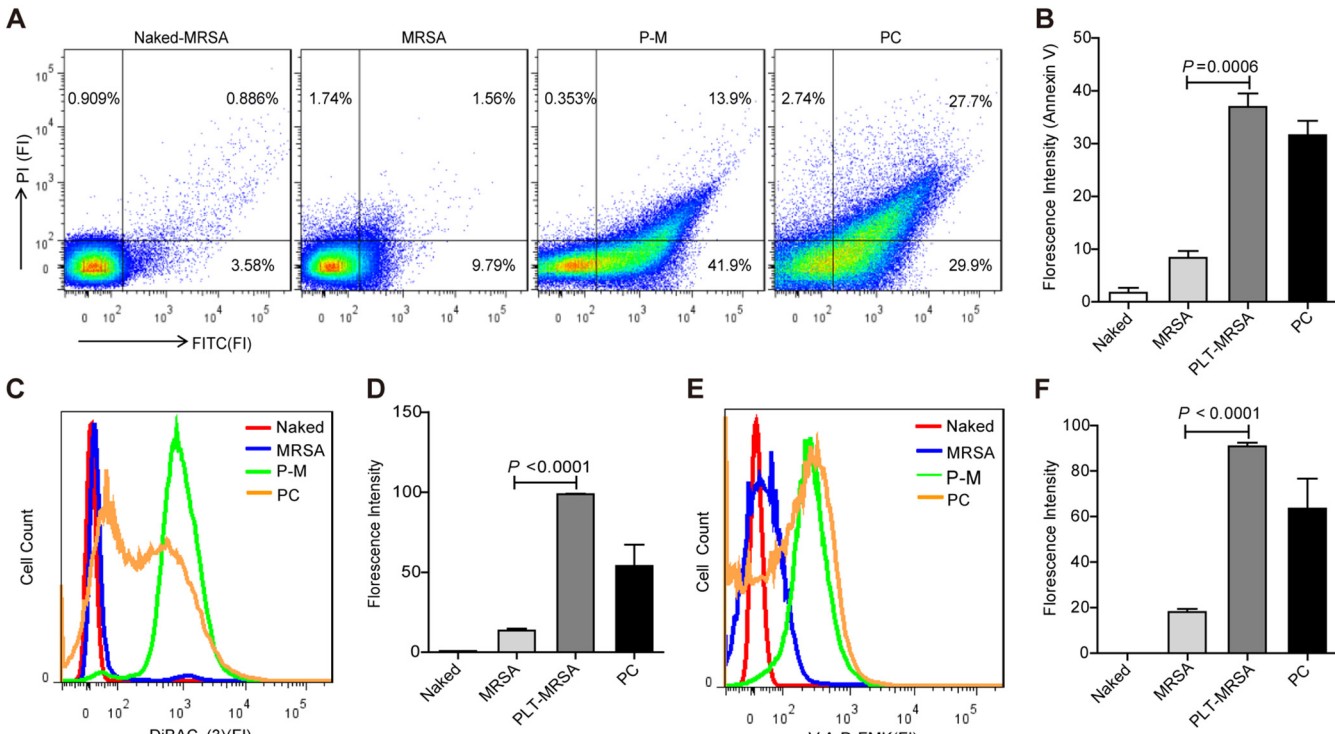

**FIG 6** Platelets-induced bacterial cell death exhibit apoptotic hallmarks. MRSA were cocultured with or without platelets for 10 h. Naked, unstained MRSA; MRSA, untreated MRSA; PLT-MRSA, platelets treated MRSA; PC, MRSA treated with MMC (5 μg/mL) for 4 h, as a positive control. (A) PS exposure detection by flow cytometry. (B) Statistical results of FITC-Annexin V fluorescence intensity in (A). (C) Detection of $DiBAC_4(3)$ labelled bacterial membrane potential by flow cytometry. (D) Statistical results of FITC-$DiBAC_4$(3) fluorescence intensity in (C). (E) Detection of V-A-d-FAM labelled intracellular caspase of bacteria by FACS analysis. (F) Statistical results of FITC-V-A-d-FAM fluorescence intensity in (E). All results have been tested at least three times. Data presented as mean ± SEM. Student's *t* test for two-group comparisons.

We evaluated DNA damage using the terminal deoxynucleotidyl transferase dUTP nick-end labelling (TUNEL) assay. Mitomycin C (MMC) has been reported to induce apoptotic-like hallmarks in model eukaryotic systems and *Escherichia coli* (38). Hence, MMC was used as the positive control. Our results showed that the fluorescence intensity of the labelled DNA fragments increased significantly after platelet treatment, suggesting that the MRSA DNA was substantially damaged (Fig. 5B and C). Next, we detected the transcriptional levels of DNA damage repair-related genes, including *recJ*, *uvrD*, *ssb*, and *dpolll*, using quantitative reverse transcription-PCR (qRT-PCR). Data showed that the expression of *recJ*, *uvrD*, and *ssb* was upregulated in platelet-treated MRSA, whereas that of *dpolll* was downregulated (Fig. 5D). Meanwhile, expression of the SOS repair-related genes *recA* and *lexA* was also upregulated after DNA damage (Fig. 5D). These results demonstrated that platelets induced extensive DNA damage in MRSA. As chromosome condensation and DNA fragmentation are hallmark features of ALD (38, 39), our data suggest that platelets induce ALD in MRSA.

Antibiotic treatment induces cell death in *E. coli*, which displays characteristic markers of apoptosis, including phosphatidylserine (PS) exposure, chromosome condensation, and DNA fragmentation (38). To verify ALD induction in MRSA after platelet treatment, we performed annexin V staining, followed by FACS, to detect PS exposure in MRSA. The percentage of annexin V positive in the platelet-treated and control MRSA was 41.9% and 9.79%, respectively (Fig. 6A and B), implying that platelet treatment induced PS exposure in MRSA. Next, we evaluated the membrane potential of platelet-treated MRSA using $DiBAC_4$(3), a lipophilic anionic fluorescent dye sensitive to the cell membrane potential (40). The fluorescence intensity of $DiBAC_4$(3) increased significantly in the PLT-MRSA group compared to that in the untreated MRSA group (Fig. 6C and D), indicating that platelet treatment depolarized the MRSA membrane. Furthermore, we detected the activity of intracellular caspase in MRSA using V-A-d-FMK. Intracellular fluorescence indicated stable binding of FITC-V-A-d-

FMK to bacterial proteins with affinity for a general caspase substrate; the increase in fluorescent intensity reflected an increase in the concentrations of these proteins (38). As shown in Fig. 6E and F, the fluorescence intensity of FITC-V-A-d-FMK was enhanced in the PLT-MRSA group, implying that the activity of intracellular caspase increased in MRSA after platelet treatment. Collectively, these data suggest that platelets induce ALD in MRSA. Furthermore, platelet lysate treatment induced apoptotic changes in MRSA via depolarizing the membrane (Fig. S6A and B) and increasing the intracellular caspase activity (Fig. S6C and D).

**Quenching of hydroxyl radicals prevents the platelet-induced ALD of MRSA.** Previous studies reported that AMPs, including coprisin, psacotheasin, and arenicin-1, promote the production of excessive reactive oxygen species (ROS) in *Candida albicans* (41–43). Notably, OH•, which is an important regulator of apoptosis, activates several apoptosis indicators. This suggests that AMPs trigger apoptosis by inducing ROS overproduction. In our study, platelet treatment induced OH• overproduction and ALD in MRSA. To test whether the increased OH• production observed following platelet treatment is causally related to ALD of MRSA, we added NAC and Bpy to the platelet and MRSA coculture and observed the changes in PS exposure, membrane potential, and intracellular caspase activity in MRSA. Our results showed that the addition of NAC to the coculture resulted in the inhibition of PS exposure (Fig. 7A and B), reduced membrane depolarization (Fig. 7C and D), and reduced activity of intracellular caspase (Fig. 7G and H) in MRSA. Consistently, the addition of Bpy to the coculture led to the inhibition of PS exposure (Fig. 7A and B), membrane depolarization (Fig. 7E and F), and intracellular caspase activity in MRSA (Fig. 7I and J). These results confirmed that the platelets induced ALD in MRSA by promoting OH• overproduction.

## DISCUSSION

In this study, we demonstrate that platelets can inhibit MRSA growth both *in vivo* and *in vitro*. Platelet treatment induced oxidative stress and OH• overproduction in MRSA, and the excessive OH• led to ALD in MRSA (Fig. 7K). Our findings highlight the novel function of platelets in fighting MRSA infections. Moreover, this study paves the way for future platelet transfusion therapy in the clinical setting.

We found that platelet-derived antimicrobial factors induced excessive OH• production in MRSA, which eventually resulted in ALD of MRSA. We believe in the credibility of the proposed mechanism for the following reasons. First, MRSA growth was significantly inhibited in the PLT-MRSA coculture. Second, MRSA suffered severe oxidative stress following platelet treatment, as evident by OH• overproduction. The platelet-induced MRSA death exhibited characteristic hallmarks of apoptosis, including membrane depolarization and PS exposure in the early stage, followed by increased intracellular caspase activity, DNA fragmentation, and chromatin condensation during the late stage of apoptosis. Third, addition of NAC and Bpy to the PLT-MRSA coculture weakened the bacteriostatic effect of platelets and alleviated the apoptotic changes in MRSA. Importantly, treatment with platelet lysates also caused ALD in MRSA by inducing excessive OH• production.

Several studies have demonstrated the antibacterial effects of platelets. The synergistic action of Kupfer cells and platelets facilitates rapid encapsulation of blood-borne bacteria and confers protection to the host (24, 44). Some researches have reported that after platelets interact with *S. aureus*, the $\delta$ particles of platelets release adenosine diphosphate (ADP) and adenosine triphosphate (ATP), while the $\alpha$ particles release AMPs and kinocidins. The released ADP continuously activates the surrounding resting platelets to increase the release of PMPs and kinocidins (21, 45, 46). Moreover, platelets release $\beta$-defensin 1 (HBD-1) upon stimulation with alpha-toxin. Platelet-derived HBD-1 significantly inhibits the proliferation of *S. aureus* and induces neutrophil extracellular trap formation (47). However, a few studies have explored whether platelets exert similar bacteriostatic effects on drug-resistant bacteria. To the best of our knowledge, this study is the first to provide evidence that platelets

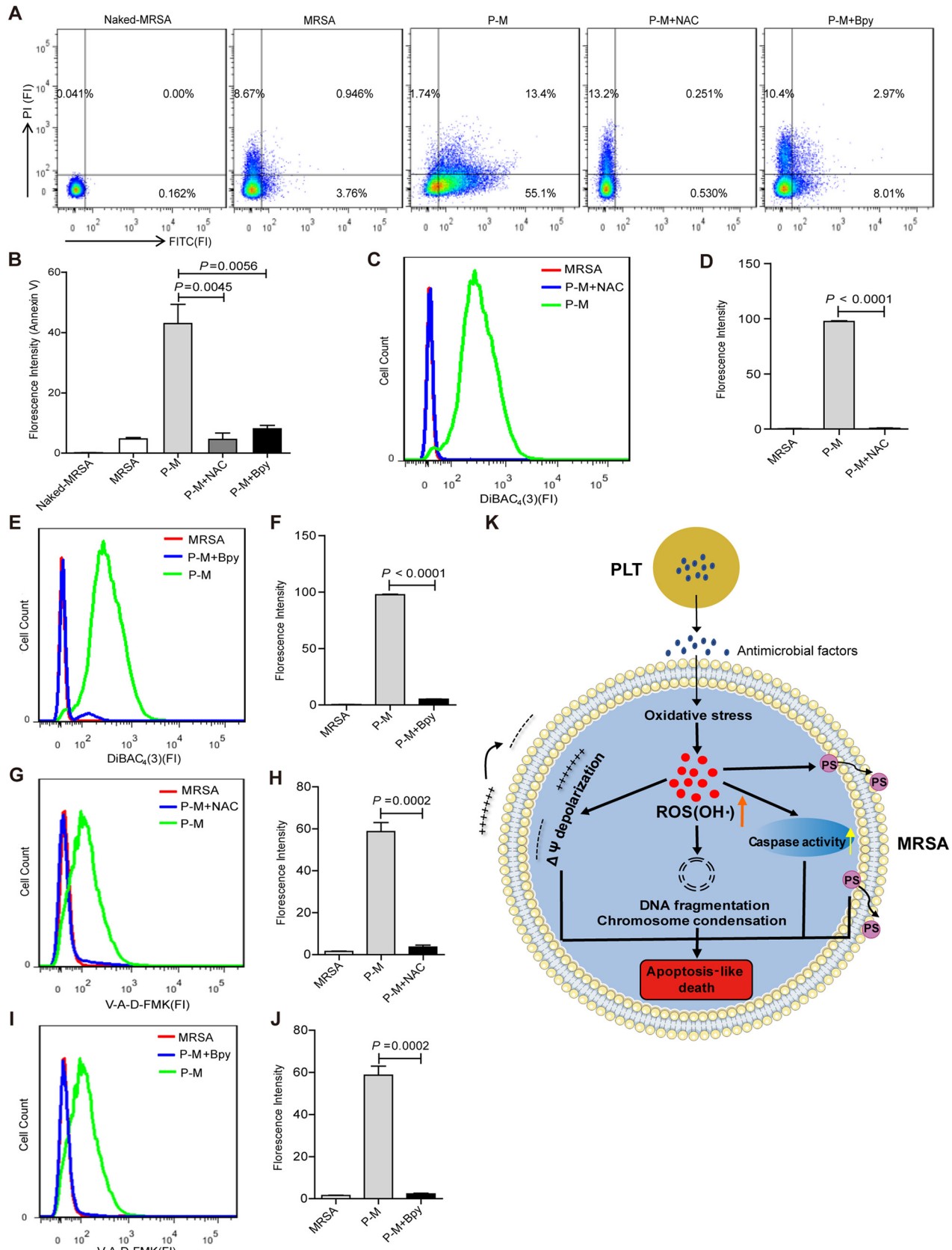

**FIG 7** Quenching of OH• prevents platelet-induced apoptosis of MRSA. MRSA were cocultured with or without platelets for 10 h, adding 6 mM NAC and 500 μM Bpy, respectively, to the coculture system. Naked-MRSA, unstained MRSA; MRSA, untreated MRSA; P-M, platelets treated MRSA;

can significantly inhibit MRSA growth *in vitro* and play an important role in improving the survival of MRSA-infected mice.

Here, platelets induced OH• overproduction in MRSA, ultimately leading to the ALD of MRSA. The pathway of OH•-mediated ALD has been previously reported. Hydroxyurea (HU) induces cell death in *E. coli* by inducing the uptake of iron and oxidative stress via excessive production of OH• through the Fenton reaction, accompanied by DNA damage responses and membrane depolarization (35). Environmental stressors, such as antibiotics and UV light, modulate the tricarboxylic acid cycle in bacteria, resulting in transient consumption of NADH and destabilization of iron and sulphur clusters. This eventually stimulates the Fenton reaction to produce excessive OH•. The highly destructive OH• not only causes severe DNA damage beyond the repair abilities of the SOS response of bacteria but also induces membrane depolarization and PS exposure, thereby leading to ALD in bacteria (34, 39, 48–51). This suggests that OH• overproduction plays a key role in the bacterial death caused by external stressors such as antibiotics. Bpy protects *E. coli* from antibacterial drugs by disrupting OH• accumulation. In addition, combined treatment with Bpy and NAC inhibits the antibacterial activities of daunomycin, moxifloxacin, and oxacillin against *S. aureus* (37). Our findings provide credible evidence that OH• plays a key role in the platelet-mediated ALD of MRSA. Collectively, our findings and the results of previous studies suggest that the mechanism underlying the bacteriostatic effect of platelets may be similar to that underlying the effects of certain antibiotics.

Although our study presents some compelling findings regarding platelet-mediated inhibition of MRSA growth, it has some limitations. Although human apheresis platelets used in the experiment were adequately washed to remove the plasma component, a little fibrinogen may still have remained in the coculture system due to its expression in the platelets. It has been reported that platelet adhesion and encapsulation of *S. aureus* is an important step in platelet sterilization (47). Platelet-derived antimicrobial peptides play a key antibacterial role after adhering to and wrapping bacteria (16, 19). Likewise, in our study, trypan blue staining results, detected using an oil microscope, showed that several dead MRSA cells were surrounded by lysed platelets (Fig. S2D). Therefore, the effect of very few fibrinogens in coculture system of platelet and MRSA was very slight. Because platelet lysates exert the same antibacterial effect as platelets, promote OH• overproduction in MRSA, and induce ALD in MRSA, it is possible that the antibacterial factors present in platelets play a key role in these anti-MRSA effects. However, platelets contain various AMPs and kinocidins, such as $\beta$-defensin 2, thymosin $\beta$4, CXCL4, CXCL7, etc., all of which exert significant antibacterial effects (16–18, 22, 23, 52). The platelet content, which plays a key role in anti-MRSA effect and induces OH• overproduction, remains indistinct, and the mechanism underlying the platelet-induced ALD in MRSA remains unknown. In future studies, we will explore the mechanism of platelet-induced ALD in MRSA. In addition, unravelling whether platelets exert antibacterial effects on other drug-resistant bacteria warrants further investigation.

In conclusion, our results provide the first evidence of the inhibitory effect of platelets on MRSA growth and demonstrate that platelets induce OH• mediated ALD in MRSA. Similar to the use of chemical methods for promoting ROS-mediated cancer cell apoptosis (53), enhancement of ROS production and promotion of bacterial apoptosis have important implications for improving antimicrobial therapies (36). Therefore, development of

**FIG 7** Legend (Continued)

P-M+NAC: NAC was added into the platelets treated MRSA system; P-M+Bpy, Bpy is added to platelets treated MRSA system. (A) The detection of PS exposure by flow cytometry. (B) Statistical results of FITC-Annexin V fluorescence intensity in (A). (C) The detection of bacterial membrane potential with DiBAC$_4$(3) by flow cytometry. (D) Statistical results of FITC-DiBAC$_4$(3) fluorescence intensity in (C). (E) Membrane potential detected by flow cytometry in each group. (F) Statistical results of fluorescence intensity in (E). (G) The detection of intracellular caspase of bacteria with V-A-d-FMK by flow cytometry. (H) Statistical results of FITC-V-A-d-FMK fluorescence intensity in (G). (I) Intracellular caspase detection by flow cytometry in each group. (J) Statistical results of fluorescence intensity in (I). (K) Model for platelets induce OH•-mediated apoptosis-like cell death in MRSA. $\Delta\Psi$, membrane potential. All results have been tested at least three times. Data presented as mean $\pm$ SEM. Student's *t* test for two-group comparisons.

novel antimicrobial agents to induce OH• mediated bacterial apoptosis will provide an alternative approach for the prevention and control of bacterial diseases and may help solve the growing problem of antibiotic resistance.

## MATERIALS AND METHODS

**Evaluation of MRSA resistance, culture, and counts.** Standard MRSA (USA300, American Type Culture Collection BAA-1717, Manassas, VA, USA) and clinically isolated MRSA strains were procured from the laboratory department of Xijing Hospital of Fourth Military Medical University. MRSA genomic DNA was extracted, and the *mecA* gene was amplified using PCR. Agarose gel electrophoresis was performed to confirm the presence of *mecA* in the standard and clinically isolated MRSA strains. In addition, methicillin and vancomycin were used to validate drug resistance of the two MRSA strains. The MICs of methicillin for MRSA standard strains (ATCC BAA-1717) and clinical isolates strains were detected by broth microdilution method in Mueller-Hinton medium. Briefly, 12 sterile test tubes were used to prepare antibacterial liquid with different concentration gradients using MH broth medium by multiple dilution method. The concentrations of antibiotics in tubes 1 to 11 were 1,280, 640, 320, 160, 80, 40, 20, 10, 5, 2.5, 1.25 $\mu$g/mL, respectively, and tube 12 was MH broth medium as control. Then, 10 $\mu$L of different concentrations of antibacterial liquid was added into the 96-well sterile plate, and 10 $\mu$L of MH broth medium was added into the 12th well as the growth control. Bacterial suspension (about $10^8$CFU/mL) with a concentration equivalent to 0.5 McFarland standards was prepared by MH broth medium 1:1,000 diluted, and 90 $\mu$L was added to each of the 12 wells. The concentrations from the first to the 11th wells were 128, 64, 32, 16, 8, 4, 2, 1, 0.5, 0.25, 0.125 $\mu$g/mL, respectively. After sealing, the bacteria were incubated 16–20 h in a 37°C shaker. The minimum drug concentration to completely inhibit the growth of bacteria was MIC, and $OD_{600}$ was determined with a Microplate Reader.

The culturing and counting of MRSA were performed as previously described (54). Briefly, MRSA was inoculated on Luria-Bertani (LB) agar plates and cultured at 37°C for 24 h to obtain a single colony. Next, we inoculated a single colony of MRSA in LB medium and cultured at 37°C for 6 h to 8 h until the logarithmic growth phase was reached. Bacterial concentration was determined at 600 nm using a spectrophotometer (UV-2550, Shimadzu, Kyoto, Japan). For the *in vitro* studies, MRSA was diluted approximately to $1 \times 10^5$ CFU/mL and incubated in LB with or without washed platelets. The bacterial suspension was diluted in each group, using the serial dilution method, every 2 h after coculture, and 20 $\mu$L bacterial suspension was inoculated in LB plates. Bacterial colonies were counted after 18 h to 24 h of incubation at 37°C. For *in vivo* studies, the MRSA culture was precipitated via centrifugation, washed with phosphate-buffered saline (PBS) three times, and suspended in PBS to the count of $1 \times 10^9$ CFU/mL. In order to ensure the accuracy of the $OD_{600}$ value and colony count results, washing was carried out prior to detection to prevent MRSA aggregation.

**Preparation of human washed platelets.** Human apheresis platelets were centrifuged at 382 × *g* for 5 min and then washed with a washing solution (PBS containing a 10% ACD solution; the ACD solution contained 1.33 g sodium citrate, 0.47 g citrate, and 3 g glucose per 100 mL). After washing gently two times (382 × *g*, 5 min), the platelets were suspended to achieve a count of 150 × $10^9$/L in LB medium. The purity of the washed platelets was determined as previously described (29). All relevant procedures were performed in accordance with the requirements of the Xijing Hospital Ethics Committee.

**Preparation of mice platelets.** C57BL/6 mice, 6 to 8 weeks old, were purchased from the Animal Center of the Fourth Military Medical University. All animal experiments in this study were conducted in accordance with the regulations for animal protection and management of the Fourth Military Medical University. Platelets from C57BL/6 mice were collected as previously described (29). Briefly, blood was collected from the hearts of C57BL/6 mice and stored in a PBS solution (containing a 20% ACD solution). Before platelet transfusion, the blood was centrifuged two times (150 × *g* each time) for 10 min, and red and white blood cells were removed. The upper phase containing platelets was transferred into a new sterile tube, centrifuged at 750 × *g* for 15 min, and platelet precipitates were collected and suspended in PBS. The number of platelets was counted using a whole blood autoanalyzer (XP-100, Sysmex, Kobe, Japan). Finally, freshly isolated platelets (200 $\mu$L, 1,200 to 1,500 × $10^9$/L) were injected into the mice through the tail veins.

**Mouse model of MRSA infection.** MRSA (200 $\mu$L, 1 × $10^9$ CFU/mL) was intravenously injected into the mice, and the blood was collected four times, at 0 h, 8 h, 18 h, and 24 h, from mice in each group through the orbital veins. The blood volume collected each time was 50 $\mu$L and was stored in a PBS solution (containing 20% ACD solution) for subsequent analysis of platelet and WBC counts using a whole blood autoanalyzer, while the number of bacteria in the peripheral blood was counted using an LB plate. Mice were subjected to general anaesthesia before each blood collection. The mice were euthanized 24 h after MRSA infection, and the livers were collected for histochemical analysis. All relevant procedures were performed in accordance with the requirements of the Xijing Hospital Ethics Committee.

**Platelet transfusion in mice.** C57BL/6 female mice (6 to 8 weeks old; approximately 20 g) were divided into the PBS, MRSA, and PLT transfusion groups (5 mice/group). In the PLT transfusion group, platelets were transfused 6 h after MRSA infection, and the blood was collected through the ophthalmic vein at 0 h, 8 h, 18 h, and 24 h. Platelets and white blood cells in the peripheral blood of mice collected at different times were assessed using a whole blood autoanalyzer, and bacterial counts were analyzed using an LB plate. The clinical scores of MRSA infection were graded as previously described (55). The mice were euthanized 24 h after MRSA infection, and the livers were collected for histochemical analysis.

**Mouse model of platelet depletion.** Platelet depletion was performed in C57BL/6 mice as previously described (29). Briefly, C57BL/6 female mice (6 to 8 weeks old, approximately 20 g) were divided

into the PBS, MRSA, and PLT depletion groups (5 mice/group). In the PLT depletion group, mice were injected with CD42b MAb (Emfret, Eibelstadt, Germany; 2 mg/kg in 200 $\mu$L sterile PBS) via the tail veins 2 h before MRSA infection. Blood was collected through the ophthalmic vein at 0 h, 16 h, and 24 h. The platelet coagulation function of the mice was assessed as previously described (29). Platelet and WBC counts in the peripheral blood of mice collected at different times were measured using a whole blood autoanalyzer, and bacterial counts in peripheral blood were assessed using an LB plate. The mice were euthanized 24 h after MRSA infection, and the livers were collected for histochemical analysis.

**Histological analysis.** Tissues were subjected to histological analysis as previously described (29). Briefly, one mouse was randomly selected from each group and euthanized 24 h after MRSA infection. The liver was obtained, preserved at $-20°C$ after OCT embedding, and sliced. Tissue sections were stained with hematoxylin and eosin and observed under a microscope (200$\times$ magnification). To analyze the ultrastructure of MRSA, MRSA samples were centrifuged (2392 $\times$ $g$, 5 min), pelleted, fixed with 3% glutaraldehyde at 4°C for at least 2 h, and then sliced for sample preparation. Sections were observed under a transmission electron microscope (JEM-1230, Jeol USA, Peabody, MA, USA), and digital images were captured with a CCD camera (Olympus, Tokyo, Japan).

**Detection of bacterial proliferation.** Briefly, the CFSE Kit (KeyGen Biotech, Nanjing, China) was used to detect MRSA proliferation. After counting, MRSA was labelled with CFSE according to the manufacturer's instructions. The labelled MRSA (1 $\times$ 10$^5$ CFU/mL) was then incubated for 10 h with or without platelets. Finally, MRSA was collected and suspended in an PBS buffer for flow cytometric analysis (BD Biosciences, USA).

**RNA analysis.** After 10 h of coculture with or without platelets, RNA was extracted from MRSA using the RNeasy RNA isolation kit (Qiagen, Dusseldorf, Germany). The expression of 11 genes related to oxidative phosphorylation and DNA damage repair, viz. *ndhO*, *ndhR*, *qoxC*, *qoxD*, *azr*, *uvrD*, *recJ*, *ssb*, *dpoIII*, *recA*, and *lexA*, was analyzed using qRT-PCR; *16S rRNA* was used as the housekeeping gene. The sequences of primers used for analyzing *uvrD*, *recJ*, *ssb*, and *dpoIII* expression have been previously reported (29). The sequences of primers used for analyzing other genes were as follows:

*16S rRNA*, F(5′–3′): ACGTGGATAACCTACCTATAAGACTGGGAT, R(5′–3′): TACCTTACCAACTAGCTA ATGCAGCG; *recA*, F(5′–3′): AATGCGCTAGGTGTAGGTGG, R(5′–3′): TCGATAAATGCTGCCACCCC; *lexA*, F(5′–3′): GCCTCAATCATACTGTCGCCT, R(5′–3′): AGCAGGTGTTCCTATTACCGC; *qoxC*, F(5′–3′): CATCCGCTATAC ACCATCCCT, R(5′–3′): TCGCTAGGTATCGTTTGGGC; *qoxD*, F(5-3′): TAAGTGTGAAGAGTGACCGCC, R(5′–3′): TTGGCTTTGCATTCGTCCAAG; *ndhO*, F(5′–3′): TGCGACCATGTTTCTTTGCG, R(5′–3′): GCCACGCTTACCATT TGACC; *ndhR*, F(5′–3′): CGTCGTTAATGCAGGCTGATG, R(5′–3′): CACGAAACGCATTGACTGGA; *azr*, F(5′–3′): TTCAGTTTTATGCGGTTCGGC, R(5′–3′): TGCTGAAGGACCACAAGGTTT.

**Detection of the formation of hydroxyl radicals.** HPF (Sigma-Aldrich, St. Louis, MO, USA), which emits fluorescence after reacting with OH$^\bullet$, was used to detect OH$^\bullet$ levels. Briefly, MRSA was cocultured with or without platelets for 10 h and washed thrice with sterile PBS. The samples were then diluted at 1:100 in PBS (pH 7.2). Next, HPF was added to the diluted samples (final concentration: 10 $\mu$M) and incubated for 1 h at 25°C in the dark. After incubation, the samples were washed two times with PBS (pH 7.2) and suspended in 500 $\mu$L PBS. Finally, the fluorescence intensity of FITC-HPF was analyzed using flow cytometry.

**Analysis of DNA fragmentation.** MRSA samples were collected after 10 h of coculture with or without platelets. DNA fragmentation was detected using the TUNEL assay (In Situ Cell Death Detection Kit, Fluorescein, Roche, Mannheim, Germany). Samples were prepared according to the manufacturer's instructions and analyzed using flow cytometry.

**Detection of membrane depolarization using FACS.** DiBAC$_4$(3) (Sigma-Aldrich, St. Louis, MO, USA) was used to detect the membrane potential of MRSA. MRSA cells were cultured and washed as described above. DiBAC$_4$(3) (0.1 $\mu$g/$\mu$L; 1 $\mu$L) was added to 1 mL diluted cells for staining. The cells were then incubated at 25°C for 30 min in the dark. After staining, the samples were pelleted, the supernatant removed, and the cells suspended in 500 $\mu$L PBS. Finally, the samples were analyzed for the fluorescence intensity of FITC-DiBAC$_4$(3) using flow cytometry.

**Analysis of phosphatidylserine exposure.** MRSA was cultured and washed as described above. The samples were diluted to 1:100 in a 1$\times$ binding buffer. For staining, Annexin V and PI (2.5 $\mu$L each) (FITC Annexin V Apoptosis Detection Kit I, BD Biosciences, USA) were added to 100-$\mu$L diluted cells and incubated for 30 min at 25°C in the dark. After incubation, the samples were pelleted, the supernatant removed, and the cells suspended in 500 $\mu$L of 1$\times$ binding buffer. Finally, the fluorescent intensity was analyzed using flow cytometry.

**Detection of bacterial proteins with caspase-like substrate affinity.** We used an FITC-conjugated pan-caspase inhibitor peptide, Z-V-d-FMK (Intracellular Caspase Detection ApoStat, R&D Systems, USA), to detect whether platelets induce the expression of bacterial proteins that can bind to caspase substrate peptides. MRSA cells were cultured and washed as described above. To stain the cells, 10 $\mu$L Z-V-d-FMK-FITC was added to 1-mL diluted cells and incubated for 30 min at 37°C in the dark. Next, the cells were washed once with PBS (pH 7.2), to remove the unbound reagents, and then suspended in 500 $\mu$L PBS. Finally, the fluorescence intensity of FITC-Z-V-d-FMK was analyzed using flow cytometry.

**Statistical analysis.** Statistical significance between the means of each group was evaluated using Student's $t$ test (Prism v.5.01, GraphPad Software, La Jolla, CA, USA). The results are expressed as mean $\pm$ SEM. Statistical significance was set at $P < 0.05$.

## SUPPLEMENTAL MATERIAL

Supplemental material is available online only.

**SUPPLEMENTAL FILE 1**, PDF file, 0.8 MB.

## ACKNOWLEDGMENTS

We thank the Clinical laboratory of Xijing Hospital and the Department of Microbiology and Pathogen Biology, Fourth Military Medical University of China for providing technical support for this study. This research was supported by the National Natural Science Foundation of China (grant no. 82170226).

We declare no conflicts of interest.

E.L., Y.C., and J. Xu performed most of the experiments; S.G., N.A., J. Xin, W.W., and Z.L. contributed to the assay and discussed the results; Q.A. and J.Y. helped in analyzing the data and writing the manuscript; and W.Y. designed this study and interpreted the data.

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
