## [Reviewer comments · Microbiology Spectrum]

Microbiology Spectrum

Platelets Inhibit Methicillin-resistant *Staphylococcus aureus* by Inducing Hydroxyl Radical-mediated Apoptosis-like Cell Death

Wen Yin, Erxiong Liu, Yutong Chen, Jinmei Xu, Shunli Gu, Ning An, Jiajia Xin, Wenting Wang, Zhixin Liu, Qunxing An, and Jing Yi

Corresponding Author(s): Wen Yin, Fourth Military Medical University/XiJing Hospital

Review Timeline:

Submission Date:	November 30, 2021
Editorial Decision:	February 16, 2022
Revision Received:	April 19, 2022
Accepted:	May 19, 2022

Editor: Angela Bordin

Reviewer(s): The reviewers have opted to remain anonymous.

Transaction Report:

DOI: <https://doi.org/10.1128/spectrum.02441-21>

February 16, 2022

Prof. Wen Yin
Fourth Military Medical University/XiJing Hospital
Xi'an
China

Re: Spectrum02441-21 (Platelets Inhibit Methicillin-Resistant *Staphylococcus aureus* by Inducing Hydroxyl Radical-Mediated Apoptosis-like Cell Death)

Dear Prof. Wen Yin:

Link Not Available

Sincerely,

Angela Bordin

Journals Department
Reviewer comments:

Reviewer #1 (Comments for the Author):

The present manuscript investigates the magnitude and mechanisms of antibacterial activity of mice platelets on methicillin-resistant *Staphylococcus aureus*, in vitro and in vivo. The authors previously reported antibacterial activity of mice platelets in *S. aureus*. The aim of the present manuscript is to confirm the platelet's activity in specific *S. aureus* strains that are methicillin resistant (MRSA) and to investigate further how this activity is mediated. As the platelet's activity on bacteria and on *S. aureus* in particular was not expected to be influenced by the presence of the *mec* genetic element, it is not surprising that it was reported in MRSA just as well as in other *S. aureus* strains. The novelty of the manuscript lies more in the additional experimental evidence provided, suggesting that production of hydroxyl radicals is a key mechanism in the platelets' bactericidal activity

against *S. aureus*.

The manuscript is well written and the data presented support the conclusions. Some sections of the methods are either lacking or unclear, for example the phenotypical determination of methicillin susceptibility and the mice infection and blood collection protocols.

Specific comments:

Introduction, lines 60-62: there is some confusion in this sentence as the human immunodeficiency virus is included among major pathogenic bacteria

Introduction, lines 62-63: the percentage of MRSA among clinical *S. aureus* isolates varies dramatically in different regions of the world. The 20-50% range reported should be intended as an approximation for most, but not all, countries.

Results, lines 98-105, 111-115, and Figure 1: It is not clear how methicillin resistance was confirmed in the study strains, and how CFU counts were estimated over the course of the survival curves. Phenotypic resistance (MIC of ceftazidime in Mueller-Hinton medium or ceftazidime disc diffusion diameter) should be reported for each strain used. The antibiotic concentration should also be indicated in mg/L. Additionally, the killing/survival curves should show several time points during at least a 24h time span, rather than just one time point (figure 1 A, B). Similarly, time 0 values should be shown for panels F and H. Finally, log₁₀ values in the 12 to 13 range for the CFU/ml seem too high (panel H), and do not seem to match with the OD₆₀₀ values (panel F). Similar considerations apply to figure S1.

Results, lines 133-136, and figure 2: It would be useful to know if steps were employed to ensure possible aggregates of MRSA were separated before counting (washing, addition of antiaggregating agent, etc.). OD depends more on the number of the number of floating particles, than on their size.

Methods, lines 378-383: it is not clear how many times and how much blood was collected from mice: was each mouse sampled four times in the span of 24h, or were different mice sampled at different time points? What volume of blood was collected each time? Was local anesthesia performed? This should be described in better detail, including also adherence with the author's Institution laboratory animals guidelines.

Reviewer #2 (Comments for the Author):

See attached file

Staff Comments:

Preparing Revision Guidelines

Please return the manuscript within 60 days; if you cannot complete the modification within this time period, please contact me. If you do not wish to modify the manuscript and prefer to submit it to another journal, please notify me of your decision immediately so that the manuscript may be formally withdrawn from consideration by Microbiology Spectrum.

Spectrum 02441-21 Review

Platelets Inhibit Methicillin-Resistant *Staphylococcus aureus* by Inducing Hydroxyl Radical-Mediated Apoptosis-like Cell Death

Summary: This manuscript builds on previous work with *S. aureus* to describe how MRSA is targeted by hydroxyl radicals and subsequent apoptosis-like cell death by platelets. The authors use purified platelets and platelet lysates to show that platelets can kill *S. aureus*. They use markers for bacterial membrane potential and apoptosis, to show that these are the mechanisms behind MRSA killing by platelets.

Major Comments:

1. The reasoning behind why the authors decided to do these studies doesn't make sense. They say that they did studies with *S. aureus* and showed killing mediated by platelets, so they wanted to know if MRSA does it too. Is there reason to believe the *sccmec* element would alter this phenotype? Why would it be any different?
2. Neutrophils, macrophages and other immune cells also produce hydroxyl radicals, but are not able to eliminate *S. aureus*. (Or MRSA) So why are only platelets so efficient at doing this? Is the concentration of OH released by platelets higher? If so this should be discussed.
3. The authors would benefit from showing the OH phenotype when *S. aureus* is incubated with OH radicals from a chemical source (reagent).
4. Why isn't USA300, the most well-known MRSA strain used here as a reference? What are these strains? Are there significant genomic differences between these strains and USA300 or Newman?
5. 'Dead' MRSA classified in Figure S1F-g don't make sense. How do the authors know the bacteria are dead? Optical opacity?
6. When the authors describe a severe decrease in OD for strains incubated with platelets, how do they differentiate bacterial death from aggregation? (Similar to the decrease in OD found during agglutination).
7. Is there fibrinogen and/or fibrin in these extracts/platelet supernatants? If so this should be taken into account when measuring phenotypes. SA is known to aggregate with platelets, via ClfA and FnbpA/B
8. There are some important controls missing. For example, in Figure 6, B-F there need to be positive controls for membrane disruption and caspase activity, without which these data don't make sense.
9. The animal model with platelet transfusion is a little confusing. If 10^9 platelets were injected into a host it is likely that there will be infiltration of some immune cells! Also, there's not much of an increase in platelets either?
10. The studies with anti-CD42b treated mice cannot be taken into consideration without the appropriate controls. It would seem that the previously used model of platelet transfusion would be a perfect control to show the complementation of phenotypes.
11. Is it just OH? How are the other ROS species ruled out?

12. Why are the axes in the FloJo images in Figure 6 removed? And once again, there needs to be a positive control for PS exposure. How about E. coli! (Since they mention it).

Minor Comments

- There are numerous typos and grammatical errors that need to be fixed.
- It is a far reach to show these results and claim that the manuscript provides evidence for novel antibacterial agents or treatments against drug resistant bacteria. Authors should refrain from making these statements. They are unnecessary and not supported by the data presented.
- It should be specified whether the t-tests are paired or unpaired and whether all results were analyzed with t-tests. Also if t-tests are used then specific P values should be provided.

List of Responses

The reviewers' comments concerning our manuscript, titled "**Platelets Inhibit Methicillin-resistant *Staphylococcus aureus* by Inducing Hydroxyl Radical-mediated Apoptosis-like Cell Death (Spectrum 02441-21)**," have helped us in improving the quality of this manuscript. We have gone through all the comments carefully and tried our best to incorporate the changes recommended. All the revised parts are marked in yellow in the revised manuscript. Point-to-point responses to the reviewer's comments are given below.

1. Introduction, lines 60-62: *there is some confusion in this sentence as the human immunodeficiency virus is included among major pathogenic bacteria.*

Response: We have amended this sentence as: MRSA ranks first among the three major pathogenic microorganisms prevalent worldwide (*Mycobacterium tuberculosis*, human immunodeficiency virus, and MRSA) and accounts for 20–50% of clinically isolated *S. aureus* in many regions of the world. The revised sentence has been added in page 4, lines 59-62.

2. Introduction, lines 62-63: *the percentage of MRSA among clinical *S. aureus* isolates varies dramatically in different regions of the world. The 20-50% range reported should be intended as an approximation for most, but not all, countries.*

Response: We have revised this sentence as "MRSA ranks first among the three major pathogenic microorganisms prevalent worldwide (*Mycobacterium tuberculosis*, human immunodeficiency virus, and MRSA) and accounts for 20–50% of clinically isolated *S. aureus* in many regions of the world". The new statement has been added in page 4, lines 59-62.

3 Results, lines 98-105, 111-115, and Figure 1:

3.1 *It is not clear how methicillin resistance was confirmed in the study strains, and how CFU counts were estimated over the course of the survival curves. Phenotypic resistance (MIC of cefoxitin in Mueller-Hinton medium or cefoxitin disc diffusion diameter) should be reported for each strain used. The antibiotic concentration should also be indicated in mg/L.*

Response: (1) In the original manuscript, we stated that we performed *mecA* gene analysis via agarose gel nucleic acid electrophoresis (Fig. S1A) and detected the antibacterial effect of methicillin and vancomycin on MRSA standard strains (ATCC BAA-1717) (Fig. S1B) and clinical isolate strains (Fig. S1C), respectively. In this modified version, we detected the MICs of methicillin for MRSA standard strains (ATCC BAA-1717) (Fig. S1D to F) and clinical isolates strains (Fig. S1G to I) in Mueller-Hinton medium. The MICs of methicillin for MRSA standard strains (Fig. S1D and E) and clinical isolates strains (Fig. S1G and H) were 128 and 32 $\mu\text{g/mL}$, respectively. In addition, the concentration unit of antibiotics has been modified to mg/L . The revised statements have been added to page 18-19, lines 363-377; page 32-33, lines 708-709; supplemental material: page 2, lines 11-17 and page 3, lines 28-29.

(2) Experimental methods for estimating bacterial colony counts (CFU/mL) are as follows: MRSA was diluted approximately to 1×10^5 CFU/mL and incubated in LB with or without washed platelets. The bacterial suspension was diluted in each group, using the serial dilution method, every 2 h after co-culture, and 20 μL bacterial suspension was inoculated in LB plates. Bacterial colonies were counted after 18–24 h of incubation at 37°C . This detailed description has been added to page 19, lines 383-387 of the Methods section.

3.2 Additionally, the killing/survival curves should show several time points during at least a 24h time span, rather than just one time point (figure 1 A, B). Similarly, time 0 values should be shown for panels F and H.

Response: We observed that the turbidity of the PLT-MRSA group (indicating MRSA growth) was much lower than that of the MRSA group. The optical density at 600 nm (OD_{600}) was detected after co-culture for 10 h, and we found that OD_{600} decreased significantly, indicating that the platelets had achieved an excellent bacteriostatic effect at 10 h. These results are depicted in Fig. 1A and B. Then, we established the survival curves of platelet-inhibiting MRSA by measuring the OD_{600} value and colony counts using plates (CFU/mL) every 2 h for a 24 h period. The survival curves are shown in Fig. 1E and G. Additionally, the OD_{600} values at 0 h has been added to Fig. 1F and H.

3.3 Finally, log₁₀ values in the 12 to 13 range for the CFU/ml seem too high (panel H), and do not seem to match with the OD₆₀₀ values (panel F). Similar considerations apply to figure S1.

Response: To confirm bacterial colony count results, we performed colony counts of the bacterial suspension at each time point again; results showed that the maximum colony counts (CFU/ml) of log₁₀ were between 10 and 11 (Fig. 1H, S2B and S2C).

4. Results, lines 133-136, and figure 2: It would be useful to know if steps were employed to ensure possible aggregates of MRSA were separated before counting (washing, addition of antiaggregating agent, etc.). OD depends more on the number of the number of floating particles, than on their size.

Response: In order to ensure the accuracy of the OD₆₀₀ value and colony count results, washing was carried out prior to detection to prevent MRSA aggregation. This information has been added to page 19, lines 389-391 of the Methods section.

5. Methods, lines 378-383: it is not clear how many times and how much blood was collected from mice: was each mouse sampled four times in the span of 24h, or were different mice sampled at different time points? What volume of blood was collected each time? Was local anesthesia performed? This should be described in better detail, including also adherence with the author's Institution laboratory animals guidelines.

Response: Our blood collection protocol for mice is as follows: blood was collected four times, at 0, 8, 18, 24 h, from mice in each group through the orbital veins. The blood volume collected each time was 50 µL and was stored in a 200 µL PBS solution (containing a 20% ACD solution) for subsequent analysis of platelet and WBC counts using a whole blood autoanalyzer, while the number of bacteria in the peripheral blood was counted using an LB plate. Mice were subjected to general anesthesia before each blood collection. All relevant procedures were performed in accordance with the requirements of the Xijing Hospital Ethics Committee. This detailed description has been added to page 21, lines 414-422 of the Methods section.

List of Responses

The reviewers' comments concerning our manuscript, titled "**Platelets Inhibit Methicillin-resistant *Staphylococcus aureus* by Inducing Hydroxyl Radical-mediated Apoptosis-like Cell Death (Spectrum 02441-21)**," have helped in improving the quality of the manuscript. We have gone through all the comments carefully and tried our best to incorporate all the recommended changes. All the revised sections are marked in yellow in the revised version; the revised figures will be uploaded subsequently. The point-by-point responses to the reviewer's comments are as follows:

Major Comments:

1. The reasoning behind why the authors decided to do these studies doesn't make sense. They say that they did studies with *S. aureus* and showed killing mediated by platelets, so they wanted to know if MRSA does it too. Is there reason to believe the *sccmec* element would alter this phenotype? Why would it be any different?

Response: In our previous studies, we found that platelets directly regulate DNA damage and the division of *S. aureus* (1). However, it remained unclear whether platelets have the same antibacterial effect on drug-resistant strains. MRSA is one of the most common multidrug-resistant bacteria, which poses a significant threat to human health (2, 3). It is, therefore, very meaningful to study the antibacterial effect of platelets on MRSA. In terms of the mechanism of platelet antibacterial action, different from our previous studies in *S. aureus*, this study found that platelet treatment induce severe oxidative stress and OH• overproduction in MRSA, and ultimately the excessive OH• lead to apoptosis-like death of MRSA. The revised statements have been added to page 2, lines 25-26; page 5, lines 86-90. Moreover, as shown in the figure 1 below, we found that same concentration of platelets showed an improved antibacterial effect against MRSA compared to that against *S. aureus*.

Figure 1 Platelets inhibit MRSA and *S. aureus* growth *in vitro*. (A) The photos of MRSA co-cultured with or without platelets. (B) Comparison of OD₆₀₀ of two group. (C) The photos of *S. aureus* co-cultured with or without platelets. (D) Comparison of OD₆₀₀ of two group. SA: *S. aureus*.

2. Neutrophils, macrophages and other immune cells also produce hydroxyl radicals, but are not able to eliminate *S. aureus*. (Or MRSA) So why are only platelets so efficient at doing this? Is the concentration of OH released by platelets higher? If so this should be discussed.

Response: In this study, we demonstrated that the platelets co-cultured with MRSA led to excessive generation of OH• in MRSA itself, which led to apoptosis-like death of MRSA. We therefore primarily detected the OH• produced by MRSA, rather than that produced from the platelet secretion. The exclusion of the possible interference of OH• from the platelet secretion was mainly based on the following evidence: **a.** The results shown in Fig. S4A, S4B, and Fig. 4A provide evidence that MRSA suffered severe oxidative stress after platelet treatment. **b.** In order to inhibit the ability of platelets to produce OH•, platelet lysates were prepared in advance and then applied to MRSA. Results showed that the increased levels of OH• in MRSA were consistent with those in the platelets-treated group (Fig. S5F and G). **c.** In the revised manuscript, mitomycin C (MMC) has been mentioned as the positive control. Results showed that the intracellular OH• levels were also significantly increased after the treatment of MMC (5μg/mL) with MRSA for 4 h (Fig. 4B and C).

3. The authors would benefit from showing the OH phenotype when *S. aureus* is incubated with OH radicals from a chemical source (reagent).

Response: We conducted experiments by including H₂O₂ (10 mM) into the culture medium of MRSA, and the results showed that H₂O₂ significantly inhibited the growth of MRSA (Fig. S2E and F).

4. Why isn't USA300, the most well-known MRSA strain used here as a reference? What are these strains? Are there significant genomic differences between these strains and USA300 or Newman?

Response: MRSA standard strains (ATCC BAA-1717) used in our experiments are indeed USA300 strains (USA300-HOU-MR/*Staphylococcus aureus* subsp.//*aureus* Rosenbach). The purchase information is shown in the figure 2 below. This information has been added to page 6, line 103; page 18, line 357.

第一条商品货号、品名、数量、单价				
	编号 Item	描述 Description	数量 Qty.	单价 Price
1	BNCC 278066	ATCC BAA-1717/TCH1516 [USA300-HOU-MR]/ Staphylococcus aureus subsp.// aureus Rosenbach	1	7000
2	BNCC 280286	ATCC 25904/Newman D2C [NCTC 10833]/ Staphylococcus aureus subsp.// aureus Rosenbach	1	7000
包装邮寄费 Handing Charge				免
合计 Total				¥14000
第二条质量与验收标准				

Figure 2 The purchase information of MRSA standard strains (USA300, ATCC BAA-1717).

5. 'Dead' MRSA classified in Figure S1F-g don't make sense. How do the authors know the bacteria are dead? Optical opacity?

Response: In this revised manuscript, we added the bacterial colony count after platelet treatment; result showed that the bacterial colony count of MRSA in the platelet-treated group was significantly reduced (Fig. S2C). In addition, detection of trypan blue staining using an oil microscope showed that several dead MRSA were surrounded by lysed platelets (Fig. S2D). These results indicate that platelet treatment caused MRSA death.

6. When the authors describe a severe decrease in OD for strains incubated with platelets, how do they differentiate bacterial death from aggregation? (Similar to the decrease in OD found during agglutination).

Response: In order to ensure the accuracy of the OD₆₀₀ value, washing was carried out before detection to prevent MRSA aggregation. We performed bacterial colony counting by plate to demonstrate the antibacterial effect of platelets (Fig. 1H and S2C). This experimental detail has been added to page 19, lines 389-391 of the Method section.

7. Is there fibrinogen and/or fibrin in these extracts/platelet supernatants? If so this should be taken into account when measuring phenotypes. SA is known to aggregate with platelets, via ClfA and FnbpA/B.

Response: Although human apheresis platelets used in the experiment were adequately washed to remove the plasma component, a little fibrinogen may still be have remained in the co-culture system due to its expression in the platelets. It has been reported that platelet adhesion and encapsulation of *S. aureus* is an important step in platelet sterilization (4). Platelet-derived antimicrobial peptides play a key antibacterial role after adhering to and wrapping bacteria (5, 6). Likewise, in our study, trypan blue staining results, detected using an oil microscope, showed that several dead MRSA were surrounded by lysed platelets (Fig. S2D). Taken together, the bacterial colony count and trypan blue staining results illustrated the antibacterial effect of platelets on MRSA (Fig. 1H, S2C and S2D). This information has been added to the discussion part in page 16-17, lines 325-335.

8. There are some important controls missing. For example, in Figure 6, B-F there needs to be positive controls for membrane disruption and caspase activity, without which these data don't make sense.

Response: Mitomycin C (MMC) has been reported to induce apoptotic-like hallmarks in model eukaryotic systems and *E. coli* (7). This detailed description has been added to page 11, lines 214-216. Hence, MMC was used as the positive control. The related indicators of apoptosis, including DNA fragmentation (Fig. 5B and C), phosphatidylserine exposure (Fig. 6A and B), membrane potential depolarization (Fig. 6C and D), and increased intracellular caspase activity of bacteria (Fig. 6E and F), were detected after MRSA was treated with MMC (5 μ g/mL) for 4 h.

9. The animal model with platelet transfusion is a little confusing. If 10^9 platelets were injected into a host it is likely that there will be infiltration of some immune cells! Also, there's not much of an increase in platelets either?

Response: **a.** In our study, blood collected from C57BL/6 mice was centrifuged twice (150 \times g each time) for 10 min, and red and white blood cells were removed before platelet transfusion. A third centrifugation step (750 \times g, 15 min) was performed to obtain platelet precipitates, which were then suspended in PBS. The detailed

preparation of mouse platelets is described in page 20, lines 405-411. After three centrifugation steps, most white and red blood cells were removed, and platelet suspension cell count results showed that the white blood cell count was not detected (Fig. S3G). These findings indicate that the effect of very few immune cells on platelet transfusion was very slight.

b. Platelet infusion in mice was performed after 6 h of MRSA infection, and the earliest time point for blood collection from mice was 8 h after infection. Blood was collected from the mice 2 h after platelet infusion (Fig. 2A to E). It has been reported that platelets serve as "sentinels" for tissue damage and microbial invasion, monitor the integrity of blood vessels, and help the body build an effective immune system (8). During infection, platelets quickly travel to the site of infection in large numbers (9). This detailed description has been added to page 4, lines 68-70. Thus, in this experiment, as the mice were in a state of acute infection, transfused platelets may quickly reach the infected site after platelet transfusion. As a result, the number of platelets in the peripheral blood did not increase significantly two hours later (Fig. 2C), but the infection status of the mice was alleviated (Fig. 2F to H).

10. *The studies with anti-CD42b treated mice cannot be taken into consideration without the appropriate controls. It would seem that the previously used model of platelet transfusion would be a perfect control to show the complementation of phenotypes.*

Response: To verify that platelets play a certain antimicrobial role *in vivo*, we conducted two experiments—platelet transfusion to elevate platelet levels and anti-CD42b antibody treatment to deplete platelets. This two-experiment design was used as a parallel control to support the antibacterial effect of platelets in mice. The results are shown in Figures 2 and 3.

11. *Is it just OH? How are the other ROS species ruled out?*

Response: In general, ROS consist of three main types: superoxide (O_2^-), hydrogen (H_2O_2), and hydroxyl radicals (OH^\bullet) (10). As shown in figures 3 and 4 below, when cells are subjected to oxidative stress, O_2^- produced by the oxidative respiration chain is converted into H_2O_2 under the action of superoxide dismutase. H_2O_2 in the presence of Fe^{2+} generates OH^\bullet through the Fenton reaction (10-12). So, both O_2^- and H_2O_2 can

eventually be converted into $\text{OH}\cdot$ *in vivo*. $\text{OH}\cdot$ is considered the most toxic and deadly of the three types. As shown in Figure 5 below, the highly destructive $\text{OH}\cdot$ acts as an "executor" of cell death, which directly damages DNA, lipids, and proteins, and ultimately leads to cell death (13). Therefore, we took the most important $\text{OH}\cdot$ into account. The new statement has been added to page 9-10, lines 174-183.

Figure 3 Generation of ROS during antibiotic-induced stress conditions (10).

Figure 4 Schematic Representation of Cellular Response to HU Treatment (11). These effects disrupt respiratory chain activity, causing an increase in superoxide production, eventually leading to increased OH• production and cell death.

Figure 5 Mechanisms Leading to ROS-Mediated Antibiotic Killing (13).

12. Why are the axes in the FloJo images in Figure 6 removed? And once again, there needs to be a positive control for PS exposure. How about *E. coli*! (Since they mention it).

Response: In the revised manuscript, we have added the axes to images in Figures 6 and 7. In addition, we used the MMC and MRSA co-culture system as a positive control to detect MRSA-related indicators of apoptosis, including DNA fragmentation (Fig. 5B and C), phosphatidylserine exposure (Fig. 6A and B), membrane potential depolarization (Fig. 6C and D), and increased intracellular caspase activity of bacteria (Fig. 6E and F). Considering that the antibacterial effects of platelets on MRSA and *E. coli* may be different in our preliminary study, *E. coli* was not used as a positive control. As shown in the figure 6 below, platelets have no similar antibacterial effect on *E. coli* compared to MRSA.

Figure 6 Platelets inhibit MRSA and *E. coli* growth *in vitro*. (A) The photos of MRSA co-cultured with or without platelets. (B) Comparison of OD₆₀₀ of two group. (C) The photos of *E. coli* co-cultured with or without platelets. (D) Comparison of OD₆₀₀ of two group.

Minor Comments

1. *There are numerous typos and grammatical errors that need to be fixed.*

Response: We have corrected all the errors in the manuscript.

2. It is a far reach to show these results and claim that the manuscript provides evidence for novel antibacterial agents or treatments against drug resistant bacteria. Authors should refrain from making these statements. They are unnecessary and not supported by the data presented.

Response: We have carefully revised the mentioned expression in our manuscript. The new statement has been revised in page 3, lines 45-46; page 5, lines 97-98; page 14, lines 274-276.

3. It should be specified whether the t-tests are paired or unpaired and whether all results were analyzed with t-tests. Also if t-tests are used then specific P values should be provided.

Response: In this study, an unpaired t-test was used to analyze all the results, and specific *P*-values have been provided in the revised manuscript.

REFERENCES

1. Xu J, Yi J, Zhang H, Feng F, Gu S, Weng L, Zhang J, Chen Y, An N, Liu Z, An Q, Yin W, Hu X. 2018. Platelets directly regulate DNA damage and division of *Staphylococcus aureus*. *Faseb j* 32:3707-3716.
2. Mitcheltree MJ, Pisipati A, Syroegin EA, Silvestre KJ, Klepacki D, Mason JD, Terwilliger DW, Testolin G, Pote AR, Wu KJY, Ladley RP, Chatman K, Mankin AS, Polikanov YS, Myers AG. 2021. A synthetic antibiotic class overcoming bacterial multidrug resistance. *Nature* 599:507-512.
3. Willyard C. 2017. The drug-resistant bacteria that pose the greatest health threats. *Nature* 543:15.
4. Kraemer BF, Campbell RA, Schwertz H, Cody MJ, Franks Z, Tolley ND, Kahr WH, Lindemann S, Seizer P, Yost CC, Zimmerman GA, Weyrich AS. 2011. Novel anti-bacterial activities of β -defensin 1 in human platelets: suppression of pathogen growth and signaling of neutrophil extracellular trap formation. *PLoS Pathog* 7:e1002355.
5. Blair P, Flaumenhaft R. 2009. Platelet alpha-granules: basic biology and clinical correlates. *Blood Rev* 23:177-89.

6. Yeaman MR. 2010. Bacterial-platelet interactions: virulence meets host defense. *Future Microbiol* 5:471-506.
7. Dwyer DJ, Camacho DM, Kohanski MA, Callura JM, Collins JJ. 2012. Antibiotic-induced bacterial cell death exhibits physiological and biochemical hallmarks of apoptosis. *Mol Cell* 46:561-72.
8. Gaertner F, Massberg S. 2019. Patrolling the vascular borders: platelets in immunity to infection and cancer. *Nat Rev Immunol* 19:747-760.
9. Cloutier N, Allaeyes I, Marcoux G, Machlus KR, Mailhot B, Zufferey A, Levesque T, Becker Y, Tessandier N, Melki I, Zhi H, Poirier G, Rondina MT, Italiano JE, Flamand L, McKenzie SE, Cote F, Nieswandt B, Khan WI, Flick MJ, Newman PJ, Lacroix S, Fortin PR, Boilard E. 2018. Platelets release pathogenic serotonin and return to circulation after immune complex-mediated sequestration. *Proc Natl Acad Sci U S A* 115:E1550-e1559.
10. Dwyer DJ, Kohanski MA, Collins JJ. 2009. Role of reactive oxygen species in antibiotic action and resistance. *Curr Opin Microbiol* 12:482-9.
11. Davies BW, Kohanski MA, Simmons LA, Winkler JA, Collins JJ, Walker GC. 2009. Hydroxyurea induces hydroxyl radical-mediated cell death in *Escherichia coli*. *Mol Cell* 36:845-60.
12. Kohanski MA, Dwyer DJ, Hayete B, Lawrence CA, Collins JJ. 2007. A common mechanism of cellular death induced by bactericidal antibiotics. *Cell* 130:797-810.
13. Van Acker H, Coenye T. 2017. The Role of Reactive Oxygen Species in Antibiotic-Mediated Killing of Bacteria. *Trends Microbiol* 25:456-466.

May 19, 2022

Prof. Wen Yin
Fourth Military Medical University/XiJing Hospital
Xi'an
China

Re: Spectrum02441-21R1 (Platelets Inhibit Methicillin-resistant *Staphylococcus aureus* by Inducing Hydroxyl Radical-mediated Apoptosis-like Cell Death)

Dear Prof. Wen Yin:

Your manuscript has been accepted, and I am forwarding it to the ASM Journals Department for publication. You will be notified when your proofs are ready to be viewed.

A condition of publication in any ASM Journal is that authors make data fully available, without restriction, except in rare circumstances; therefore, a data availability statement should be added at the end of the Material and Methods section (Production staff will assist with this procedure). The minimum data set for which authors are required to provide access includes all data, metadata, and methods used to reach the conclusions in the submitted paper and any additional data required to replicate the study findings. ASM recommends that authors make the data available for sharing through a suitable public repository; alternatively, authors may upload data, particularly small data sets, as Supplemental Material. Please verify all links to sequence records, if present, and make sure that each number retrieves the full record of the data. If a new accession number is not linked or a link is broken, provide production staff with the correct URL for the record. If the accession numbers for new data are not publicly accessible before the expected online posting of the article, publication of your article may be delayed; please contact the ASM production staff immediately with the expected release date.

Sincerely,

Angela Bordin
Editor, Microbiology Spectrum
